# Schistosomes Impede ATP-Induced T Cell Apoptosis In Vitro: The Role of Ectoenzyme SmNPP5

**DOI:** 10.3390/pathogens11020155

**Published:** 2022-01-26

**Authors:** Catherine S. Nation, Akram A. Da’dara, Manal Elzoheiry, Patrick J. Skelly

**Affiliations:** Molecular Helminthology Laboratory, Department of Infectious Disease and Global Health, Cummings School of Veterinary Medicine, Tufts University, North Grafton, MA 01536, USA; catherine.nation@tufts.edu (C.S.N.); Akram.Da_darah@tufts.edu (A.A.D.); dr_manal@mans.edu.eg (M.E.)

**Keywords:** *Schistosoma*, adenosine triphosphate, immunomodulation, Tregs, purinergic signaling

## Abstract

Schistosomes (blood flukes) can survive in the bloodstream of their hosts for many years. We hypothesize that proteins on their host-interactive surface impinge on host biochemistry to help ensure their long-term survival. Here, we focus on a surface ectoenzyme of *Schistosoma mansoni*, designated SmNPP5. This ~53 kDa glycoprotein is a nucleotide pyrophosphatase/phosphodiesterase that has been previously shown to: (1) cleave adenosine diphosphate (ADP) and block platelet aggregation; and (2) cleave nicotinamide adenine dinucleotide (NAD) and block NAD-induced T cell apoptosis in vitro. T cell apoptosis can additionally be driven by extracellular adenosine triphosphate (ATP). In this work, we show that adult *S. mansoni* parasites can inhibit this process. Further, we demonstrate that recombinant SmNPP5 alone can both cleave ATP and impede ATP-induced T cell killing. As immunomodulatory regulatory T cells (Tregs) are especially prone to the induction of these apoptotic pathways, we hypothesize that the schistosome cleavage of both NAD and ATP promotes Treg survival and this helps to create a less immunologically hostile environment for the worms in vivo.

## 1. Introduction

Human schistosomiasis is a disease caused by parasitic platyhelminths of the genus *Schistosoma*; largely, *S. mansoni*, *S. haematobium*, or *S. japonicum*. The disease can be highly debilitating, afflicting > 200 million people globally and with over 800 million people living at risk of infection [1,2]. There is no vaccine to prevent schistosome infection. Current treatment largely relies on the use of a single drug, praziquantel, and there is an urgent need for the development of new therapies to treat schistosomiasis.

People become infected when they are exposed to larval parasites called cercariae that emerge from an infected intermediate, a freshwater snail host. A cercaria can penetrate the skin of a definitive mammalian host and differentiate into a juvenile form called a schistosomulum. Schistosomula enter the vasculature and migrate through the heart and lungs to reach the hepatic portal vein where they mature. Male and female worms pair and then migrate together through the blood to the mesenteric veins (for *S. mansoni* and *S. japonicum*) or the vesical venous plexus of the bladder (for *S. haematobium*) where egg-laying begins [3]. Female worms can lay hundreds of eggs each day.

Adult worms can survive for many years in their hosts where they modulate the vascular environment and the immune response [4,5,6,7]. Our laboratory focuses on the nature of the host-interactive proteins expressed at the schistosome surface that mediate these effects [4,8,9,10,11]. Among the surface-exposed proteins of *Schistosoma mansoni* is the tegumental enzyme, SmNPP5 (GenBank accession number EU769293). SmNPP5 is a ~53 kDa glycoprotein belonging to the ectonucleotide phosphodiesterase/pyrophosphatase (E-NPP) family of enzymes [12,13]. *S. mansoni* parasites whose SmNPP5 gene has been suppressed using RNA interference are greatly diminished in their ability to infect experimental animals [12]. From this outcome, we infer that SmNPP5 fulfills an essential function for the worms; once that function is diminished by experimental manipulation (such as RNAi), the worms cannot survive.

SmNPP5 has been identified by proteomic analyses of the schistosome tegumentome [14,15]. Immunolocalization experiments and analyses of tegument surface membrane preparations confirm the protein to be tegumental [12,13]. Real-time RT-qPCR analysis showes that SmNPP5 gene expression is rapidly induced following infection of the final host; i.e., in the transition from free-living cercaria to parasitic schistosomulum and adult worm stages [12,13].

Live schistosomula and adult worms can hydrolyze the synthetic phosphodiesterase-specific substrate *p*-nitrophenyl 5′-thymidine monophosphate (*p*-Nph-5′-TMP) [12]. This ability by the worms is significantly diminished following SmNPP5 gene suppression [12]. Biochemical analysis suggests that essentially all active SmNPP5 in schistosomes is at the parasite/host interface [12]. From this, we conclude that the vital role of the enzyme involves interacting with key intravascular components of the host. What might these be?

To investigate this question, we first generated and purified functionally active, recombinant SmNPP5 using a CHO cell expression system. Recombinant SmNPP5 was shown to cleave *p*-Nph-5′-TMP in a reaction that required divalent ions and at a pH optimum of 9 [16]. In order to assess whether SmNPP5 could play a role in the known ability of schistosomes to impede blood coagulation, the protein was added to whole blood and platelet function was monitored using multiple electrode aggregometry (MEA) [16]. These experiments revealed that SmNPP5 had a profoundly inhibitory effect on platelet aggregation in a dose-dependent manner [16]. The degree of inhibition of rSmNPP5-driven aggregation was substantially higher when ADP was added to the assay as an agonist to trigger platelet activation versus when collagen was used as an agonist. These results led to the hypothesis that SmNPP5 mediates its inhibitory effects by cleaving ADP. In vitro analysis confirmed that rSmNPP5 could hydrolyze ADP to liberate Pi with a Km of 246 ± 34 µM [16]. Heat-inactivated rSmNPP5 no longer cleaved ADP nor could it inhibit platelet aggregation. From this work, we concluded that SmNPP5 expressed at the schistosome surface could degrade extracellular ADP in the local intravascular environment of the worm and, in this manner, the protein could act as an anti-coagulant. By helping to impede blood clot formation around the worms in vivo, this could allow the parasites freer movement within the vasculature.

In addition to cleaving ADP, we have also shown that recombinant SmNPP5 (as well as adult schistosomes) can cleave another exogenous purinergic signaling molecule, nicotinamide adenine dinucleotide (NAD) [17]. This is noteworthy because NAD has been reported to have a profound effect on the fate of T lymphocytes and can induce T cell apoptosis [18]. The phenomenon of NAD-induced cell death (NICD) occurs when a cell-surface ADP ribosyl transferase (ART2) hydrolyzes NAD and ribosylates the ligand-gated cation-selective channel, P2X7 (P2X7R) [19]. This induces several cellular responses; there is an influx of calcium and sodium ions, phosphatidylserine is externalized, and non-selective membrane pores form, leading to cell death [20]. As T regulatory cells (Tregs, CD4+ CD25+ FoxP3+ cells) characteristically increase the expression of P2X7R, this makes them especially prone to NICD [18]. We have shown that worms, through SmNPP5, can cleave NAD and prevent NICD in vitro [17]. We hypothesize that the schistosome cleavage of NAD in this manner promotes Treg survival to create a more immunologically hospitable environment for the worms in vivo.

In addition to acting through NAD, a second metabolite, adenosine triphosphate (ATP), can additionally activate P2X7R and, at a high concentration (mM range), can also induce T cell apoptosis [21]. ATP is considered to be the main physiological ligand for P2X7R. In this work, we aim to extend our analysis of how schistosomes might impact T cell fate by testing the hypothesis that the parasites can block ATP-driven T cell apoptosis. This seems likely because live worms have long been shown to be capable of hydrolyzing exogenous ATP [22,23] and an ATP diphosphohydrolase (SmATPDase1) was identified in *S. mansoni* adult tegument extracts by proteomic analysis [14,24]. Immunolocalization experiments confirmed SmATPDase1 as a schistosome tegument protein [25]. We generated a full-length recombinant form of SmATPDase1 in CHO-S cells that was enzymatically active and we confirmed that the protein could hydrolyze ATP (as well as ADP) [26]. An ability to cleave exogenous ATP by schistosomes would complement their ability to hydrolyze NAD in that both of these outcomes are predicted to boost Treg numbers. As Tregs are potent suppressors of immunity [27], promoting their survival would create a less immunologically hostile environment for the worms in the vasculature of their hosts.

In this work, we set out to formally ascertain whether schistosomes can block ATP-driven T cell apoptosis. As we have previously demonstrated that SmATPDase1 can cleave ATP, we wanted to determine here if a second ectoenzyme, SmNPP5, could also hydrolyze ATP and block ATP-driven T cell apoptosis in vitro.

## 2. Materials and Methods

### 2.1. Parasites and Mice

*Biomphalaria glabrata* snails (strain NMRI) infected with *Schistosoma mansoni* were obtained from the Schistosomiasis Resource Center (Biomedical Research Institute, BRI). Cercariae were obtained from infected snails and used to infect Swiss Webster mice (at ~100 cercariae/mouse). Seven weeks post-infection, the mice were euthanized by exposure to CO_2_. Adult male and female worms were recovered from the mice by vascular perfusion and cultured in complete Dulbecco’s modified Eagle’s (DMEM)/F12 medium supplemented with 10% heat-inactivated fetal bovine serum, 200 U/mL penicillin, 200 μg/mL streptomycin, 0.2 μM Triiodo-L-thyronine, 1 μM serotonin, 200 μM ascorbic acid, and 10 μg/mL human insulin and were maintained at 37 °C in an atmosphere of 5% CO_2_. Lymph nodes and spleens were removed from 6–8-week-old female BALB/c mice and T cells were purified as described below. All protocols involving animals were approved by the Institutional Animal Care and Use Committee (IACUC) of Tufts University and all animal work was done in the vivarium at Cummings School of Veterinary Medicine, Tufts University.

### 2.2. T Cell Purification

Pooled lymph nodes (inguinal, popliteal, cervical, axillary, and mesenteric) and spleens were dissected from mice and placed in ice-cold Hank’s Balanced Salt Solution (HBSS) containing 2% heat-inactivated fetal bovine serum (FBS). Cell suspensions were prepared by passing the tissue through a 40 µm cell strainer. Lymphocytes were collected by centrifugation at 300× *g* for 5 min at 4 °C and resuspended in RPMI medium. Splenocytes were collected by centrifugation in the same manner, resuspended in Red Blood Cell Lysing Buffer (Sigma, Allentown, PA, USA, Cat. No. R7757) for 2 min at room temperature, washed by centrifugation, and, finally, resuspended in RPMI. Cell numbers and viability were determined by diluting at a ratio of 1:10 with 2% Trypan Blue for 5 min, followed by counting on a hemocytometer.

T cells were isolated from the cell mixture using a Dynabeads Untouched Mouse T cell kit (Invitrogen, Carlsbad, CA, USA, Cat. No. 11413D) following the manufacturer’s instructions. T cell purity was verified by cell staining using phycoerythrin-conjugated anti-CD45R/B220 antibody (BD Biosciences, Woburn, MA, USA, Clone RA3-6B2, Cat. No. 553090) and FITC-conjugated anti-CD3 antibody (BD Biosciences, clone 145-2C11, Cat. No. 553061), followed by flow cytometry. T cell purity was consistently >95%.

### 2.3. Measuring the Impact of Schistosomes and rSmNPP5 in an ATP-Induced Cell Death Assay

To measure the effect of schistosomes on ATP-induced T cell killing, adult worms (6 male and female pairs in 500 µL assay buffer (50 mM Tris-HCl (pH 9.0), 120 mM NaCl, 5 mM KCl, 50 mM glucose, and 2 mM CaCl_2_)) were first incubated at 37 °C in the presence or absence of ATP (2 mM). The samples were then collected after 2 and 24 h and were added to cultures of isolated T cells (2.5 × 10^5^ cells/well) at a concentration of 100 µM ATP (based on initial ATP concentration). To measure the percentage of cells that were apoptotic after 30 min, all cells were washed once with Annexin Binding buffer (0.1 M Hepes pH 7.4, 1.4 M NaCl, 25 mM CaCl_2_), then stained with FITC-conjugated Annexin V (1 µg/mL, BD Biosciences, Cat. No. 556419) and propidium iodide (10 µg/mL, Invitrogen, Cat. No. P1304MP) in Annexin Binding buffer for 20 min at room temperature and protected from light. Cells were then washed twice with flow cytometry buffer (PBS, 1% BSA, 0.05% NaN_3_) and resuspended in 100 µL buffer before being subjected to flow cytometry using a BD Acurri C6 cytometer (BD Biosciences). Additional T cell incubations included in this experiment were: no addition of ATP (as a control) as well as incubation with medium in which worms (without ATP) were kept for 2 or 24 h (to test for the impact of any worm products on T cell viability). Data obtained using T cells isolated from lymph nodes and spleens were combined in the results.

A similar scheme to that just described was used to monitor the impact of recombinant SmNPP5 on ATP-induced T cell apoptosis. Briefly, 500 µL of 2 mM ATP in assay buffer was incubated with 1 µg purified rSmNPP5 at 37 °C. Aliquots were recovered at either 15 min, 30 min, 1 h, or 24 h later and added to purified T cells (at 100 µM, based on starting concentration), which were then processed as described above.

### 2.4. ATP Cleavage by Recombinant SmNPP5

Recombinant SmNPP5, expressed in suspension-adapted FreeStyle Chinese Hamster Ovary Cells (CHO-S), was purified from culture medium by standard immobilized metal affinity chromatography (IMAC) using HisTrap Excel columns, as previously described [16]. In initial experiments, the ability of rSmNPP5 vs. control protein (purified recombinant *S. mansoni* alkaline phosphatase, rSmAP [28]) to cleave ATP was measured by adding 0.5 µg of either recombinant enzyme to ATP (2 mM) in assay buffer (50 mM Tris-HCl (pH 9.0), 120 mM NaCl, 5 mM KCl, 50 mM glucose, and 2 mM CaCl_2_) at 37 °C. The concentration of ATP remaining after incubation for either 30 min or 2 h was determined using an ATP Colorimetric/Fluorometric Assay Kit, according to the manufacturer’s instructions (BioVision, Milpitas, CA, USA, Cat. No. K354100). To observe any ATP cleavage products generated by SmNPP5 action, the reaction mixture was analyzed using thin layer chromatography (TLC). In this experiment, 1 µg rSmNPP5 was added to 2 mM ATP in assay buffer at 37 °C for 1 h or 24 h. Reaction aliquots, as well as chemical standards (ATP, ADP, and AMP), were applied to a TLC silica gel 60 F_245_ (aluminum sheet, 20 × 20 cm, EMD Millipore, Billerica, MA.USA) and allowed to dry. Separation was achieved using a mobile phase composed of n-butanol:acetone:acetic acid (glacial):ammonia (5%):water (45:15:10:10:20) and the analytes were visualized under UV at 254 nm.

TLC analysis showed that SmNPP5 cleaved ATP to generate AMP but TLC could not detect the second predicted reaction product, pyrophosphate (PPi). Therefore, we looked directly for PPi following rSmNPP5 incubation with ATP for 1 h or 2 h using a Pyrophosphate (PPi) Assay Kit (Fluorometric/Colorimetric) following the manufacturer’s instructions (BioVision Cat. No. K568100).

The Michaelis–Menten constant (Km) for ATP hydrolysis by SmNPP5 was determined in standard assay buffer containing different concentrations (0–3.0 mM) of ATP as substrate, using the PPi detection kit noted above.

### 2.5. Statistical Analysis

Statistical analysis was carried out using GraphPad Prism 8 (GraphPad Software, San Diego, CA. USA). One-way ANOVA, with Tukey’s multiple comparison test, was used to assess apoptosis data (at each time point). Values of *p* < 0.05 were considered to be significant.

## 3. Results

### 3.1. Adult Schistosomes Prevent ATP-Induced Cell Death In Vitro

Flow cytometry was employed to assess the impact of adult worms on ATP-induced T cell death. The status of the cells used in these experiments was monitored by first staining them with FITC-conjugated Annexin V and propidium iodide. Using flow cytometry, the following four populations of cells were identified: live; early apoptotic; late apoptotic; and necrotic. Figure 1 depicts these categories in the cell status key in the lower left. In our quantitative assessments, the early apoptosis and apoptosis groups were combined to provide a measure of total apoptosis. For each treatment, this total apoptotic cell population is indicated by a blue box in Figure 1. Representative flow cytometry images are shown in Figure 1, panels A–G. Data from experimental replicates are combined and mean percent apoptotic values (±SEM) are presented in Figure 1H.

Control cell populations not exposed to ATP were largely healthy (Figure 1A) and had a low percent of apoptotic T cells (mean 13 ± 3% overall apoptotic, Figure 1H, leftmost bar). However, exposure to ATP triggered T cell apoptosis; these data are shown both in representative Figure 1B (where ATP was incubated at 37 °C for 2 h prior to being added to the T cell culture) and in representative Figure 1E (where ATP was incubated at 37 °C for 24 h prior to being added to the T cell culture). Figure 1H shows that the mean percent apoptosis increased to 67 ± 8% (for the 2 h ATP incubations) and 60 ± 5% (for the 24 h incubations). In contrast, preincubation of ATP with adult schistosomes for 2 h prior to being added to T cell cultures (ATP + worms) significantly reduced the mean percent of apoptotic T cells (28 ± 2%; Figure 1H; representative Figure 1C). In addition, preincubation of ATP with schistosome parasites for 24 h prior to being added to T cells completely abrogated the phenomenon of T cell apoptosis (13 ± 1% apoptosis; Figure 1H; representative Figure 1F; *p* < 0.001 at the 2 and 24 h time points, ANOVA). Note that control medium that contained worms (and no ATP) for 2 h (representative Figure 1D) or for 24 h (representative Figure 1G) did not by itself induce apoptosis; the mean percent total apoptosis here was similar to that of control cells (13 ± 3% after 2 h; 12 ± 2% after 24 h; +worms, Figure 1H).

### 3.2. rSmNPP5 Can Degrade ATP

In this experiment, either rSmNPP5 or a control protein (recombinant *S. mansoni* alkaline phosphatase, rSmAP), or no protein was incubated with ATP. After either 30 min or 2 h, the amount of ATP remaining was determined in each case. Figure 2A shows that only rSmNPP5 diminished ATP levels and at both time points. The control protein (rSmAP) or no protein (None) had no appreciable impact on ATP levels. To look for ATP cleavage products, rSmNPP5 (either in native or heat-inactivated (HI) form) was incubated with ATP for either 1 or 24 h and aliquots were then resolved by thin layer chromatography (TLC). Figure 2B shows a TLC silica gel sheet revealing the pattern of the sample migration. SmNPP5 cleavage of ATP is clear; the amount of starting material was diminished following a 1 h incubation and was further hydrolyzed by 24 h (blue arrowhead). This did not happen when ATP was incubated with heat-inactivated rSmNPP5 (Figure 2B, +HI SmNPP5). ATP cleavage generated a reaction product whose migration profile matched that of AMP (red arrowhead). Migration patterns of the standards ATP, ADP, and AMP are shown on the left panel of Figure 2B. TLC did not reveal the presence of the second predicted reaction product of SmNPP5-mediated cleavage of ATP, i.e., pyrophosphate (PPi). Figure 2C confirms that rSmNPP5 (but not heat-inactivated rSmNPP5) generated this product; when rSmNPP5 was incubated with ATP, PPi could be detected after 1 or 2 h incubation. The Michaelis–Menten constant of SmNPP5 for ATP was determined, as depicted in Figure 2D; Km = 217 µM ± 26. Figure 2E depicts the chemical reaction catalyzed by SmNPP5 as revealed by these experiments; the cleavage site on ATP is indicated by a dashed line and reaction products (AMP plus PPi) are illustrated.

### 3.3. SmNPP5 Prevents ATP-Induced Cell Death In Vitro

Having shown that rSmNPP5 could degrade ATP, the impact of this cleavage on ATP-induced cell death was assessed using essentially the same protocol as described earlier for whole worms. T cells were purified and then incubated with ATP (for 30 min) that either had or had not been exposed to rSmNPP5 at 37 °C for either 15 min, 30 min, 1 h, or 24 h, as indicated. The cells were then stained with FITC-conjugated Annexin V and propidium iodide (PI) and were subjected to flow cytometry to assess the percent of apoptotic T cells. Representative plots are shown in Figure 3A–F; data from experimental replicates were combined, and the mean percent overall apoptotic values (±SEM) are presented in Figure 3G. As before, the percent of apoptotic control cells, if they were not exposed to ATP, was low (~10% of the total cells; representative Figure 3A). However, following treatment with ATP, as expected, a significant percentage of T cells became apoptotic (representative image Figure 3B; 65 ± 2%; Figure 3G; *p* < 0.0001). Following incubation with rSmNPP5 for 15 min (representative Figure 3C), 30 min (Figure 3D), 1 h (Figure 3E), or 24 h (Figure 3F), the population of apoptotic cells was progressively reduced. Figure 3G shows this time-dependent diminution in the percent apoptotic T cells (*p* < 0.001 for 30 min, 1 h, and 24 h data compared with +ATP alone; Figure 3G). After 24 h incubation with rSmNPP5, only 12 ± 1% of the cells were apoptotic, similar to untreated control cells. It is clear that, as seen following worm preincubation with ATP, preincubating ATP with rSmNPP5 alone likewise negates T cell apoptosis in vitro.

## 4. Discussion

Schistosomes are long-lived intravascular worms of great global health significance. We hypothesize that one major reason for the success of these parasites is their ability to control the biochemistry of their local environment. To study this issue, we focus on understanding the molecular biology of prominent, host-exposed schistosome tegumental proteins. Some such proteins are essential for the uptake of vital metabolites such as glucose or amino acids across the tegument [29,30] or to permit water flux across the external membranes [31]. Other proteins can bind and cleave host metabolites that likely influence host immune and/or hemostatic responses. For instance, tegumental alkaline phosphatase (SmAP in *S. mansoni*) has been shown to cleave the bioactive lipid, sphingosine-1-phosphate, as well as the inorganic polymer, polyP; effects that could impinge on immune cell signaling and/or platelet function [28,32].

The work described here focuses on the interactions between schistosomes and the host-purinergic signaling molecule, ATP. ATP is present in negligible amounts (nmol/L) outside healthy tissues but can accumulate rapidly to high levels extracellularly in response to cell damage [33]. Molecules such as ATP thus signal damage, and are commonly referred to as alarmins or damage-associated molecular patterns (DAMPs) [34]. ATP release from cells can also be triggered by some proinflammatory stimuli [35]. Extracellular ATP can exert multiple effects, including promoting inflammatory cell migration, activating the NLRP3 inflammasome, inducing cytokine and chemokine release, driving oxygen and nitrogen radical generation, and potentiating intracellular pathogen killing [36]. An ability of schistosomes to regulate extracellular ATP levels could negate these effects and generate a less inflammatory environment around the worms.

ATP acts as a signaling molecule by activating purinergic P2 receptors. These are widely expressed in different tissues and are involved in innate and adaptive immune responses. Of relevance here are P2XR channels; these can be opened not only by NAD-driven ribosylation but also by the binding of ATP, allowing sodium and calcium influx as well as potassium efflux. P2X7R is considered to be an important regulator of T cell function; P2X7 receptor stimulation in microenvironments rich in extracellular ATP induces the death of T cell effector subsets such as Tregs [37]. Thus, lowering extracellular ATP levels could help modulate anti-worm immunity by influencing the fate of the T lymphocyte populations. In this work, we confirmed that live schistosomes could block ATP-driven T cell apoptosis. This effect is predicted to preferentially promote Treg cell survival and is noteworthy in the context of schistosomiasis because, in mouse models, Tregs play important roles in minimizing pathology [38,39,40]. In addition, reducing the Treg population in *S. japonicum*-infected mice (by treating the mice with an anti-CD25^+^ monoclonal antibody) reduced worm numbers compared with controls [41]; this implies that there is a protective role for Tregs in relation to schistosome survival. In addition, the efficacy of an anti-*S. japonicum* vaccine was greatly increased in mice whose Treg cell numbers were lowered [42]. However, in humans, the relationships between Treg levels and infection intensity or immunopathology are less clear [43].

Here, we confirmed that ATP could induce T cell killing but this ability was blocked following ATP preincubation with adult schistosomes for either 2 or 24 h before being added to T cells in culture. We argue that by preferentially inhibiting Treg apoptosis (via ATP cleavage), schistosomes work to maintain a more immunotolerant environment for themselves in vivo.

We have characterized a schistosome tegumental ectoenzyme, SmATPDase1, that can cleave ATP and could theoretically account for these results. In this work, we tested the hypothesis that a second schistosome tegumental ectoenzyme, SmNPP5, could additionally fulfil this role. SmNPP5 is expressed exclusively in intra-mammalian parasite life stages and highest gene expression is found in mated adult males where the protein is found predominantly in the dorsal tegument [12,13]. Current known substrates for SmNPP5 are ADP and NAD. Given that members of the NPP family from other organisms can cleave ATP [44], we examined here whether SmNPP5 likewise had this ability. We found that recombinant SmNPP5, expressed in secreted form from transfected CHO cells and purified by IMAC, was indeed able to degrade ATP. ATP cleavage generated the reaction products AMP and pyrophosphate (PPi). In comparison, ATP cleavage by the schistosome tegumental ectoenzyme SmATPDase1 generates ADP and phosphate (Pi). Both enzymes, SmNPP5 and SmATPDase1, cleave ADP to generate AMP and Pi.

In light of the ability of SmNPP5 to hydrolyze ATP, it is not surprising that preincubating ATP with rSmNPP5 before adding it to isolated T cells in culture impedes T cell killing and the longer the preincubation, the greater the impedance. Based on these findings, we hypothesize that schistosomes could use tegumental SmNPP5 to degrade exogenous ATP and thus block Treg apoptosis. However, as SmATPDase1 is found in approximately five-fold greater abundance in the adult male tegument compared with SmNPP5 [45], it seemed likely that SmATPDase1 plays the major role in this phenomenon in vivo. The Km of SmNPP5 for ATP (217 µM ± 26) is about half the value of that previously determined for SmATPDase for this substrate (400 µM ± 0.02) [26]. Future work will explore the ability of other schistosome life stages (particularly eggs) to degrade extracellular ATP. If deposited eggs also display ATPase activity, this might help impede Treg cell apoptosis in inflammatory tissues surrounding them. Finally, as it has been shown that extracellular ATP can regulate platelet function [46], it is possible that the schistosome ectoenzymes SmATPDase1 and SmNPP5 could also impinge on platelet activation and thrombus formation by degrading extracellular ATP.

In summary, the work described here shows that the *S. mansoni* virulence factor SmNPP5, previously shown to be an ADPase [16] and an NADase [17], is also an ATPase. By helping to control local extracellular ATP, NAD, and ADP levels, this multifunctional enzyme could play a key role in modulating what has become known as the purinergic halo around the worms in vivo [47]. As noted earlier, suppressing SmNPP5 gene expression led to parasite death in vivo, suggesting that an ability to control the local purinergic milieu is essential for schistosome survival. Blocking this capability by inhibiting SmNPP5 function might, therefore, lead to parasite death and so help to diminish the global burden of schistosomiasis.

## Figures and Tables

**Figure 1 pathogens-11-00155-f001:**
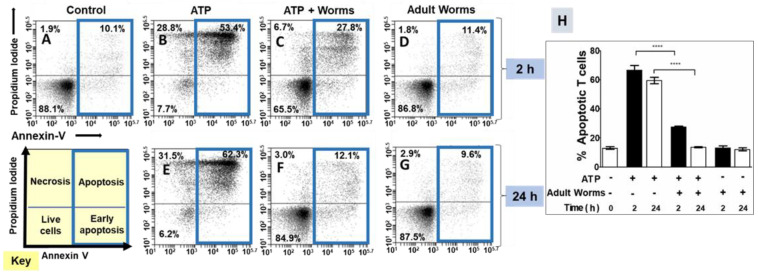
Adult worms prevent ATP-induced T cell death in vitro. Panels (**A**–**G**) are representative (from one run) flow cytometry dot plots of purified T cells following various experimental treatments (described below) and panel (**H**) (**right**) represents composite data showing the mean percent total apoptotic T cells (±SEM) following these treatments in replicate. ATP was first incubated at 37 °C for 2 or 24 h with (panels (**C**,**F**)) or without (panels (**B**,**E**)) adult schistosome parasites (male and female). An equivalent amount of ATP (100 µM, based on initial concentration) was then added to cultures of purified T cells (2.5 × 10^5^ cells/well) for 30 min. Cells were then stained and subjected to flow cytometry. Controls include T cells incubated without ATP (control, panel (**A**)) as well as cells incubated with worm culture medium that did not contain ATP (adult worms, panels (**D**,**G**)). The key at the lower left indicates cell status in each FACS plot quadrant (live, early apoptotic, apoptotic, and necrotic). Panels (**B**–**D**) show samples treated with the 2 h (ATP at 37 °C) incubation period and panels (**E**–**G**) show samples treated with the 24 h (ATP at 37 °C) incubation period. Of primary interest here are the percentages of total apoptotic cells (i.e., early + late apoptosis; right upper and lower quadrants), which are bound by blue boxes in panels (**A**–**G**). **** *p* < 0.0001 ATP vs. ATP + worms (one-way ANOVA at each time point); *n* ≥ 4 in each case.

**Figure 2 pathogens-11-00155-f002:**
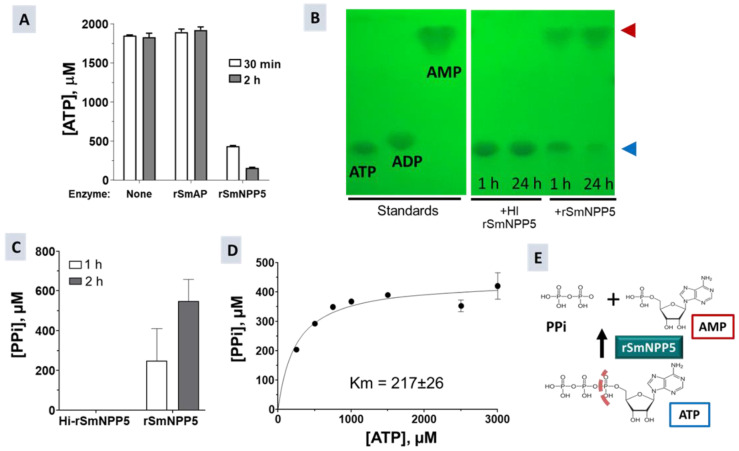
SmNPP5 cleaves ATP. (**A**) ATP (µM ± SD) remaining after either 30 min (white bars) or 2 h (grey bars) incubation with rSmNPP5, or control protein (rSmAP), or no protein (None). (**B**) Thin layer chromatography analysis of reaction products following ATP (blue arrowhead) incubation either with rSmNPP5 or with heat-inactivated (HI) rSmNPP5 for 1 h or 24 h (as indicated). Cleavage yields AMP (red arrowhead). Migration of standards is shown on the left. (**C**) Pyrophosphate (PPi, µM ± SD) generated following 1 h (white bar) or 2 h (grey bar) incubation of ATP with rSmNPP5 or with heat-inactivated (HI) rSmNPP5. (**D**) Kinetics of SmNPP5-mediated cleavage of ATP; Michaelis–Menten Km (µM ± SD) = 217 ± 26. (**E**) Depiction of ATP cleavage reaction catalyzed by SmNPP5; the structures of ATP and cleavage products AMP and PPi are shown. The dashed line indicates the site of cleavage. ATP: adenosine triphosphate; ADP: adenosine diphosphate; AMP: adenosine monophosphate; PPi: pyrophosphate.

**Figure 3 pathogens-11-00155-f003:**
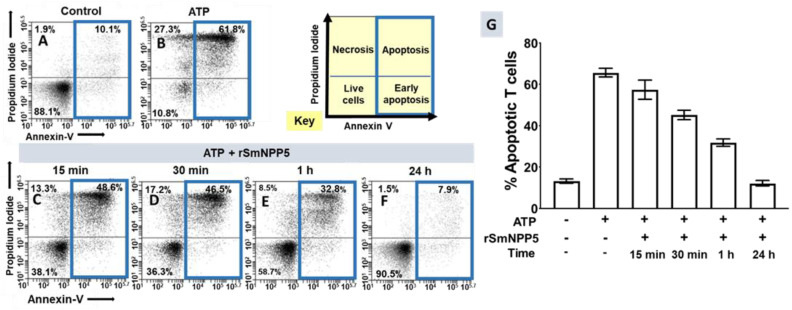
Recombinant SmNPP5 prevents ATP-induced T cell death in vitro. Panels (**A**–**F**) are representative flow cytometry dot plots of purified T cells following either exposure to ATP (panel (**B**)) or not (control, panel (**A**)), or to ATP that had been incubated with 1 µg rSmNPP5 for 15 min (panel (**C**)), 30 min (panel (**D**)), 1 h (panel (**E**)), or for 24 h (panel (**F**)). The key (yellow) indicates the cell status in each FACS plot quadrant (live, early apoptotic, apoptotic, and necrotic); the percentages of total apoptotic cells are bound by blue boxes. Panel (**G**) represents composite data from replicate experiments showing the mean percent total apoptotic T cells (±SEM) following the treatments indicated. *p* < 0.05 for 15 min data and *p* < 0.0001 for 30 min, 1 h, and 24 h data vs. + ATP control, one-way ANOVA; *n* = 6 in each case.

## Data Availability

Not applicable—all data presented herein.

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
