# Peer review of "Schistosomes Impede ATP-Induced T Cell Apoptosis In Vitro: The Role of Ectoenzyme SmNPP5"

_pathogens, 2022, doi:10.3390/pathogens11020155_

Round 1

Reviewer 1 Report

Schistosomes or blood flukes are parasitic worms, causative agents, found worldwide, of a highly debilitating disease, the schistosomiasis, touching multiple organs in infected individuals.

The article presented by Cathering S Nation et al. describes the role of the SmNPP5 enzyme on the tegument of schistosomes that might cleaves extracellular ATP, a damaged associated molecular pattern molecule, present in high concentration following cell damages. The authors hypothesize that by reducing ATP concentration via SmNPP5 activity, schistosomes prevent T cell apoptosis, in particular Treg which are prone to ATP-induced apoptosis via P2X7R, therefore establishing a more favorable immune environment for the worm survival.

Although the article is well written and provides clear and sufficient background with well detailed figures, the introduction is quite lengthy. In addition, existing treatments and the need to improve upon current strategies could be emphasized.

Figure 1 demonstrates that preincubating worms with ATP abolishes ATP-induced apoptosis of T cells. Figures 2 and 3 use recombinant smNPP5 to demonstrate the cleavage of ATP in subproducts that are ADP, AMP and PPiand its effect on T cell apoptosis.

It would be interesting if possible to show the same time points in fig 2A and Fig 2C (either 30min and 2h; or 1h and 2h in both figures) as it appears that 30min of reaction cleaves about 3.5x ATP which doubles in 2h (Fig 2A) while the amount of PPi produced doubles between 1h to 2h of incubation.

Blocking SmNPP5 effect to ascertain its role on T cell apoptosis would be interesting in similar experiments as in Fig 1 using worms and T cells in presence of SmPP5 and/or SmATPDase1 inhibitors to show its actual contribution compared with SmATPDase1 as eluded by the authors in discussion, as well as worm viability following its blockade.

In addition, the overall results lead the authors to hypothesize a role on Treg, more prone to ATP-induced apoptosis. The demonstration of protection of Treg cells from apoptosis when pre-incubated with either worms or recombinant SmNPP5 would strengthen this hypothesis.

Author Response

Reviewer 1.

The article presented by Cathering S Nation et al. describes the role of the SmNPP5 enzyme on the tegument of schistosomes that might cleaves extracellular ATP, a damaged associated molecular pattern molecule, present in high concentration following cell damages. The authors hypothesize that by reducing ATP concs3eentration via SmNPP5 activity, schistosomes prevent T cell apoptosis, in particular Treg which are prone to ATP-induced apoptosis via P2X7R, therefore establishing a more favorable immune environment for the worm survival.

Although the article is well written and provides clear and sufficient background with well detailed figures, the introduction is quite lengthy. In addition, existing treatments and the need to improve upon current strategies could be emphasized.

Thanks to the reviewer for the constructive comments about our manuscript.

As suggested by the reviewer, we have shortened the introduction and added text regarding existing treatments and the unquestionable need for new, improved anti-schistosomiasis approaches (revised manuscript, lines 34-36).

Figure 1 demonstrates that preincubating worms with ATP abolishes ATP-induced apoptosis of T cells. Figures 2 and 3 use recombinant smNPP5 to demonstrate the cleavage of ATP in subproducts that are ADP, AMP and PPiand its effect on T cell apoptosis.

It would be interesting if possible to show the same time points in fig 2A and Fig 2C (either 30min and 2h; or 1h and 2h in both figures) as it appears that 30min of reaction cleaves about 3.5x ATP which doubles in 2h (Fig 2A) while the amount of PPi produced doubles between 1h to 2h of incubation.

While we agree that the data could look better if the same time points had been chosen, the figure, as is, makes our central point that SmNPP-5 cleaves ATP to generate AMP and PPi very clearly.

Blocking SmNPP5 effect to ascertain its role on T cell apoptosis would be interesting in similar experiments as in Fig 1 using worms and T cells in presence of SmPP5 and/or SmATPDase1 inhibitors to show its actual contribution compared with SmATPDase1 as eluded by the authors in discussion, as well as worm viability following its blockade.

The reviewer makes a valuable point and, if we obtain SmNPP5 and SmATPDase1 inhibitors, it will be addressed in future work.

In addition, the overall results lead the authors to hypothesize a role on Treg, more prone to ATP-induced apoptosis. The demonstration of protection of Treg cells from apoptosis when pre-incubated with either worms or recombinant SmNPP5 would strengthen this hypothesis.

The reviewer highlights future work that could strengthen our hypothesis and we are grateful for guidance with regard to the suggested experiments.

Reviewer 2 Report

In this study, the authors demonstrated that SmNPP5, an ectoenzyme at the schistosome tegument, were able to degrade ATP and impede ATP-induced T cell apoptosis. Their technique appears sound and data presented in the manuscript seem reliable and informative to the readers of the journal. Therefore, I believe that this manuscript deserves publication. However one big question remains. Why ATP has to be degraded at the surface of the adult worms, not by eggs, in spite of the minimal concentration in the blood stream ? If deposited eggs have ATPase activities, it would be more efficient to impede T cell apoptosis in the inflammatory tissues and regional lymphatic tissues. Please discuss this point in the Discussion section.

Minor points;

2.2. Measuring the impact of schistosomes and rSmNPP5 in an ATP-induced cell death assay (page 3)

Although they start this section with ‘To measure the effect of ATP on T cell survival’, the purpose of these experiments was to measure the effect of adult schistosomes on the ATP-induced T cell apoptosis.

3.1. Adult schistosomes prevent ATP-induced cell death in vitro (page 5)

The authors state; ‘preincubation of ATP with schistosome parasites for 24 hours prior to being added to T cells completely abrogated the ability of ATP to induce T cell apoptosis’. This sentence should be something like; ‘preincubation of ATP with schistosome parasites for 24 hours prior to being added to T cells completely inhibited the T cell apoptosis’. Because schistosomes degraded ATP, but did not impede the ability of ATP to induce T cell apoptosis.

#3 Michaelis-Menton, not Michaelis–Menten, appears a couple of times in the text and a figure legend.

Author Response

Reviewer 2.

In this study, the authors demonstrated that SmNPP5, an ectoenzyme at the schistosome tegument, were able to degrade ATP and impede ATP-induced T cell apoptosis. Their technique appears sound and data presented in the manuscript seem reliable and informative to the readers of the journal. Therefore, I believe that this manuscript deserves publication. However one big question remains. Why ATP has to be degraded at the surface of the adult worms, not by eggs, in spite of the minimal concentration in the blood stream ? If deposited eggs have ATPase activities, it would be more efficient to impede T cell apoptosis in the inflammatory tissues and regional lymphatic tissues. Please discuss this point in the Discussion section.

Thanks to this reviewer for the overall positive comments regarding this paper.

The reviewer makes the valid point that an ability by parasite eggs to degrade ATP could be helpful and, while we have not looked at this issue in the current paper, we do discuss this point in the revised Discussion, as suggested (revised manuscript, lines 339-341).

Minor points;

2.2. Measuring the impact of schistosomes and rSmNPP5 in an ATP-induced cell death assay (page 3)

Although they start this section with ‘To measure the effect of ATP on T cell survival’, the purpose of these experiments was to measure the effect of adult schistosomes on the ATP-induced T cell apoptosis.

The reviewer is correct, and we have amended the text here (line 150) to make clear that the purpose of the experiment noted was to measure the effect of adult schistosomes on ATP-induced T cell killing.

3.1. Adult schistosomes prevent ATP-induced cell death in vitro (page 5)

The authors state; ‘preincubation of ATP with schistosome parasites for 24 hours prior to being added to T cells completely abrogated the ability of ATP to induce T cell apoptosis’. This sentence should be something like; ‘preincubation of ATP with schistosome parasites for 24 hours prior to being added to T cells completely inhibited the T cell apoptosis’. Because schistosomes degraded ATP, but did not impede the ability of ATP to induce T cell apoptosis.

Once again, we have altered the text here to make our intentions clearer, as recommended by the reviewer (line 226).

#3 Michaelis-Menton, not Michaelis–Menten, appears a couple of times in the text and a figure legend.

We have corrected the Michaelis-Menten spelling throughout, with thanks to the reviewer for pointing out our error.

Reviewer 3 Report

Overall impression:

This is another example of an excellent functional study of tegument proteins in schistosomes from Dr. Skelly's group. While I do have some minor questions regarding techniques and concerns about the focus on T cells in the discussion, I have no major concerns regarding this manuscript. Specifically, I do not feel any additional experiments are required to make this suitable for publication in Pathogens.

Major  points:

None noted.

Minor points:

General: Please include reference numbers for reagents such as the antibodies/FITC-conjugated Annexin V and ATP/Colorimetric/Fluorometric Assay Kit. While not strictly necessary, I personally appreciate it as it helps me ensure that I use reagents consistent with the rest of the field.

Regarding the T cell survival assays (materials and methods 2.2): I am curious as to the impact of culturing parasites in the assay buffer for extended periods of time. The pH of the buffer is relatively high (pH 9.0 according to materials and methods) and therefore is not physiological. Additionally, prolonged culture at a relatively high pH may injure the parasites and induce other changes that don't reflect physiological conditions inside the host. Is there a reason that the assay isn't carried out at pH ~7.40 and/or in parasite culture media? It might be a good idea to perform this experiment in more physiological conditions, but I am not an expert with this assay so I appreciate there may be technical justifications for the conditions used that I am unaware of.

Regarding style: Bolding of figure callouts (e.g. "Figure 1B" and "Figure 1E" in the second paragraph on page 5) is not consistent throughout the manuscript.

Regarding the discussion: My understanding is that the extracellular concentration of ATP is relatively low outside of a handful of cases (reference 34) which includes platelet degranulation. While the author's results do show in vitro inhibition of ATP-induced T cell apoptosis, I wonder if another role might be regulation of hemostasis. I think it is prudent to mention other possible functions of SmNPP5-mediated ATP hydrolysis in the discussion outside of inflammation/T cells.

Please include a citation for the concept of the "purinergic halo" (page 8, last paragraph).

Other thoughts:

An interesting experiment, albeit outside of the scope of the present study, would be to examine T cell apoptosis in the circulating T cells/thymus of infected vs. uninfected mice to see if perhaps schistosomes are capable of globally regulating T cell survival. Though SmNPP5 is a tegument surface enzyme, it is possible that shed tegument membrane and/or extracellular vesicles could allow SmNPP5 (as well as other surface enzymes) to act globally throughout the host. In vivo functional studies of these enzymes would require non-trivial RNAi/transplant experiments, but could yield exciting insight into the mechanisms by which schistosomes influence host biology.

Author Response

Reviewer 3.

This is another example of an excellent functional study of tegument proteins in schistosomes from Dr. Skelly's group. While I do have some minor questions regarding techniques and concerns about the focus on T cells in the discussion, I have no major concerns regarding this manuscript. Specifically, I do not feel any additional experiments are required to make this suitable for publication in Pathogens.

We thank the reviewer for the very encouraging words about our work.

Major  points: None noted.

Minor points:

General: Please include reference numbers for reagents such as the antibodies/FITC-conjugated Annexin V and ATP/Colorimetric/Fluorometric Assay Kit. While not strictly necessary, I personally appreciate it as it helps me ensure that I use reagents consistent with the rest of the field.

We have added reference/catalog numbers for the reagents listed.

Regarding the T cell survival assays (materials and methods 2.2): I am curious as to the impact of culturing parasites in the assay buffer for extended periods of time. The pH of the buffer is relatively high (pH 9.0 according to materials and methods) and therefore is not physiological. Additionally, prolonged culture at a relatively high pH may injure the parasites and induce other changes that don't reflect physiological conditions inside the host. Is there a reason that the assay isn't carried out at pH ~7.40 and/or in parasite culture media? It might be a good idea to perform this experiment in more physiological conditions, but I am not an expert with this assay so I appreciate there may be technical justifications for the conditions used that I am unaware of.

Over the time period of the experiments described here, despite the relatively high pH, the worms appear healthy. Our reason for choosing pH 9 in this work is because this is the optimal pH of the enzyme under study, SmNPP5.

Regarding style: Bolding of figure callouts (e.g. "Figure 1B" and "Figure 1E" in the second paragraph on page 5) is not consistent throughout the manuscript.

We have made the bolded text consistent throughout, as suggested by the reviewer.

Regarding the discussion: My understanding is that the extracellular concentration of ATP is relatively low outside of a handful of cases (reference 34) which includes platelet degranulation. While the author's results do show in vitro inhibition of ATP-induced T cell apoptosis, I wonder if another role might be regulation of hemostasis. I think it is prudent to mention other possible functions of SmNPP5-mediated ATP hydrolysis in the discussion outside of inflammation/T cells.

We thank the reviewer for the observation regarding ATP and platelet degranulation, and we have added text to the revised work pointing this out (revised manuscript, lines 342-344).

Please include a citation for the concept of the "purinergic halo" (page 8, last paragraph).

A reference (#47) regarding the term “purinergic halo” has been added, as recommended.

Other thoughts:

An interesting experiment, albeit outside of the scope of the present study, would be to examine T cell apoptosis in the circulating T cells/thymus of infected vs. uninfected mice to see if perhaps schistosomes are capable of globally regulating T cell survival. Though SmNPP5 is a tegument surface enzyme, it is possible that shed tegument membrane and/or extracellular vesicles could allow SmNPP5 (as well as other surface enzymes) to act globally throughout the host. In vivo functional studies of these enzymes would require non-trivial RNAi/transplant experiments, but could yield exciting insight into the mechanisms by which schistosomes influence host biology.

We thank the reviewer for these useful and practical ideas regarding future work.